# Psychomotor Speed and Fall Risk in Older Adults with Sarcopenia and Frailty: A Cross-Sectional Study

**DOI:** 10.3390/medicina61040706

**Published:** 2025-04-11

**Authors:** Justina Kilaitė, Rūta Dadelienė, Valentina Ginevičienė, Erinija Pranckevičienė, Asta Mastavičiūtė, Ieva Eglė Jamontaitė, Alina Urnikytė, Ildus I. Ahmetov, Vidmantas Alekna

**Affiliations:** 1Clinic of Internal Diseases and Family Medicine, Institute of Clinical Medicine, Faculty of Medicine, Vilnius University, 01513 Vilnius, Lithuania; 2Faculty of Medicine, Vilnius University, 01513 Vilnius, Lithuania; ruta.dadeliene@mf.vu.lt (R.D.); valentina.gineviciene@mf.vu.lt (V.G.); erinija.pranckeviciene@mf.vu.lt (E.P.); asta.mastaviciute@mf.vu.lt (A.M.); ieva.jamontaite@mf.vu.lt (I.E.J.); vidmantas.alekna@mf.vu.lt (V.A.); 3Faculty of Informatics, Vytautas Magnus University, 44248 Kaunas, Lithuania; 4Translational Health Research Institute, Faculty of Medicine, Vilnius University, 01513 Vilnius, Lithuania; alina.urnikyte@mf.vu.lt; 5Sport and Exercise Sciences, Faculty of Science, Liverpool John Moores University, Liverpool L3 5AF, UK

**Keywords:** falls, psychomotor speed, older adults, sarcopenia, frailty

## Abstract

*Background and Objectives*: Information on how psychomotor speed is associated with the risk of falling is scarce, even in older adults. Therefore, this study aimed to determine the relationship between falls and psychomotor speed in older adults with sarcopenia and frailty. *Materials and Methods*: A total of 204 subjects (aged 83 (77–87) years) participated in this study: 161 women (78.9%) and 43 men (21.1%). The history of falls was assessed by asking whether the subject had experienced a fall in the past 12 months. Psychomotor speed was evaluated by reaction time and frequency of movement. Sarcopenia was diagnosed based on the EWGSOP2 criteria. Frailty was confirmed if the participants met ≥3 criteria according to the Fried frailty criteria. The relationship between falls and psychomotor speed was measured using partial correlations. Binary logistic regression analysis was used to assess if psychomotor speed had an impact on falls. *Results*: Sarcopenia was confirmed in 93 (45.58%) and frailty in 91 (44.6%) subjects. Both sarcopenia and frailty were present in 62 (30.39%) participants. In the sarcopenia and frailty group, falls were related to simple reaction time (r = 0.444, *p* = 0.002), hand movement frequency in 10 s (r = −0.352, *p* = 0.014), and in 60 s (r = −0.312, *p* = 0.026). In women with sarcopenia and frailty, there were relationships between falls and simple reaction time (r = 0.68, *p* = 0.002), complex reaction time (r = 0.406, *p* = 0.004), hand movement frequency in 10 s (r = −0.614, *p* = 0.001), and in 60 s (r = −0.584, *p* = 0.001). In regression analysis, it was found that each millisecond increase in reaction time was associated with a 1.5% higher fall risk in the participants with sarcopenia (OR: 1.015 [1, 1.031], *p* = 0.048). *Conclusions*: This study demonstrates that slower psychomotor speed, particularly reaction time, is linked to a higher risk of falls in older adults with sarcopenia and frailty, especially in women.

## 1. Introduction

Sarcopenia and frailty are two major conditions affecting older adults. Sarcopenia is a disease characterized by a progressive loss of muscle function and mass [1]. Frailty is a multidimensional and dynamic condition characterized by declines in reserve and function across multiple physiological systems, where the ability to cope with everyday or acute stressors becomes compromised [2]. Both sarcopenia and frailty lack a single standard definition. It is estimated that globally, depending on the definition and cut-off values used, sarcopenia is prevalent in 10 to 27% of older adults, while frailty prevalence is around 12–24% [3,4]. Sarcopenia and frailty have many adverse health-related outcomes and are related to an increased risk of malnutrition, dementia, and postoperative complications [5,6,7]. Both sarcopenia and frailty are associated with an increased risk of falling [8,9].

Falls are one of the geriatric syndromes with a negative impact on health and a high occurrence rate. It is estimated that the prevalence of falls in older people is 26.5% worldwide [10]. This shows that falls are common in older populations. Also, falls are associated with an increased risk of injuries, fractures, fear of falling, hospitalization, and even mortality [11,12,13]. According to the World Health Organization, adults older than 60 years are at the highest risk of death or serious injury from falls, and this risk increases with age. In addition, when people fall, they suffer serious injuries such as bruises, hip fractures, or head trauma. Falls are caused by physical, sensory, and cognitive changes associated with aging, combined with an environment that is not adapted to an aging population. Moreover, older women are more likely to fall and sustain severe injuries [14]. That is why it is essential to understand what factors are associated with the risk of falls and how this could impact fall risk reduction.

Psychomotor speed is a part of our processing speed and is described as the speed at which a response is made to a given stimulus [15]. Psychomotor speed is important in helping older adults maintain everyday activities. However, the aging process affects cognitive and motor functions, and psychomotor speed tends to slow down [16,17]. Previous studies have reported that psychomotor performance correlated with basic activities of daily living, especially in the fine motor skills of lower limbs and balance areas [18].

Evidence shows that a slower cognitive processing speed is associated with an increased fall risk in older adults [19,20]. Although previous studies linked a slower processing speed with fall risk, research that specifically investigates psychomotor speed in older adults with sarcopenia and frailty remains limited. There is a lack of knowledge of psychomotor speed in older adults with sarcopenia, and frailty is more impaired than in the normal aging process. Also, there is a gap in evidence on how the psychomotor skills of the upper limbs are related to fall risk. Therefore, we hypothesize that slower simple and complex reaction times, along with reduced hand movement frequency, will be significantly associated with an increased risk of falls in older adults with sarcopenia and frailty.

## 2. Materials and Methods

Community-dwelling older adults were recruited for this cross-sectional study. The inclusion criteria were an age of 65 years or more and community-dwelling women and men. The exclusion criteria were moderate cognitive impairment—with a score of <21/30 on the mini-mental state examination (MMSE)— and acute illness. A total of 227 people were invited to participate in this study. Eighteen subjects declined the offer, and one participant was excluded due to the age criteria. There was a lack of data for sarcopenia and frailty for 4 participants. Therefore, a total of 204 subjects (aged 83 (77–87) years) participated in this study: 161 women (78.9%) and 43 men (21.1%). All participants gave their written informed consent prior to enrolment. The Lithuanian Regional Biomedical Research Ethics Committee has approved the study protocol (decision No. 2022/6-1448-918).

All subjects were measured for height (cm) and weight (kg). A history of falls was assessed by asking whether the subject had experienced a fall in the past 12 months. The number of diseases and medications taken were collected from medical records. Polypharmacy was described as taking 5 or more medications. The Short Physical Performance Battery (SPPB) was used to measure physical performance. Physical activity was assessed by the Physical Activity Scale for the Elderly (PASE). Everyday activities were evaluated by two questionnaires: Activities of Daily Living (ADL) and Instrumental Activities of Daily Living (IADL).

A physical characteristic of psychomotor speed was evaluated by reaction time and frequency of movement with a tapping test. A personal computer, a special computer program, and reactiometer RA-1 (JSC BALTEC CNC TECHNOLOGIES, Kaunas, Lithuania) were used. The reactiometer was calibrated according to the manufacturer’s instructions. Simple reaction time was measured by participants sitting in front of the reactiometer and pressing a button as soon as possible with their right hand when the light signal was on. Complex psychomotor reaction time was measured by participants sitting in front of the reactiometer and pressing a button with their right hand when the green light was on and with their left hand when the red light was on. Mistakes, if the wrong button was pressed, were counted. The tapping test was performed by asking participants to sit at the table and hold a stick. When the participants heard an auditory signal, they had to tap the board as frequently as possible. Tapping was measured for 10 and 60 s. The reaction time and tapping tests used in this study have demonstrated high test-retest reliability (ICC = 0.84, ICC = 0.98, respectively) in previous older adult populations [21,22].

Sarcopenia was diagnosed based on the criteria proposed by the European Working Group on Sarcopenia in Older People in 2018 (EWGSOP2) [23]. Sarcopenia was confirmed if low muscle strength and mass were present. Also, severe sarcopenia was diagnosed if low physical performance was assessed. Muscle mass was measured by dual-energy X-ray absorptiometry (iDXA, GE Lunar, Madison, WI, USA). Muscle strength was evaluated by handgrip strength and measured by a hydraulic dynamometer (JAMAR, Patterson Medical, Glossop, UK). Both devices—the DXA machine and hand dynamometer—were calibrated according to the manufacturer’s instructions. Physical performance was evaluated by a 4-m walking test. Frailty was confirmed based on the criteria put forward by Fried and colleagues [24]. Frailty was diagnosed if the participants met three or more of the five criteria: weight loss, weakness, exhaustion, slowness, and low physical activity. Both sarcopenia and frailty had to be present for participants to be included in the sarcopenia and frailty group.

The normality of the data was examined using the Shapiro–Wilk test. Data were not normally distributed; therefore, nonparametric tests were used in the statistical analysis. Continuous data were reported as the median and 25th–75th percentile. Nominal data were reported as frequencies (number, percentage). Differences between the groups in the univariate analysis were analyzed using the Kruskal–Wallis test. The relationship between falls and psychomotor speed was measured using partial correlations. Partial correlations were adjusted by clinical characteristics that were significantly different between groups in the univariate statistics. Binary logistic regression was used to assess a participant’s fall risk due to its ability to model dichotomous outcomes and adjust for multiple confounders. Model fitting information was evaluated by Hosmer–Lemeshow goodness-of-fit and pseudo-R2. Variables to the model were added using a backward stepwise method (Wald). The regression model was adjusted for the clinical parameters significantly differing between groups’ clinical characteristics. A power analysis was conducted using GPower (version 3.1.9.7), indicating that a minimum of 102 participants was required to detect a significant association with 80% power at α = 0.05. A significance level (*p*-value) of <0.05 was considered statistically significant. Statistical analysis was performed using IBM SPSS Statistics Windows software version 21 (IBM, Armonk, NY, USA).

## 3. Results

Out of all participants, 82 (40.2%) were allocated to the control group because they did not have frailty or sarcopenia. Sarcopenia was confirmed in 93 (45.58%) subjects, while frailty was present in 91 (44.6%) participants. Both sarcopenia and frailty were diagnosed in 62 (30.39%) participants. The basic descriptive characteristics of the study population are shown in Table 1.

Compared to the control group, participants with sarcopenia and frailty were older, had more diseases, and had a lower weight and BMI. Also, they had more daily activity impairments. Polypharmacy was present in more than half of all the study participants. In addition, subjects with sarcopenia and frailty had lower muscle strength, muscle mass, and fat mass. Furthermore, their physical performance was poorer, and they were slower than the control group. In addition, participants with sarcopenia and frailty were less physically active than the control group (see Table 1 for the PASE score).

Out of all participants, 133 (65.2%) reported at least one fall in the past 12 months. In the control group, 33 (40.2%) participants suffered a fall. In addition, more than half of all the men and women reported falls: 28 (65.1%) in the men group and 105 (65.2%) in the women group (Figure 1). Seventy-six (81.7%) participants with sarcopenia had a fall, while in the frailty group, 47 (47.5%) subjects suffered a fall. In the sarcopenia and frailty group, 56 (90.3%) participants reported at least one fall in the past year.

The falls and psychomotor speed characteristics of the study participants are shown in Table 2.

As seen in Table 2, simple and complex reaction times in participants with sarcopenia and frailty were slower. Also, the control group had faster hand movement frequency than the subjects with sarcopenia and frailty. When comparing the reaction time and hand movement frequency between men and women, it was found that women had slower hand movement frequency.

Multiple associations were found between falls and psychomotor speed (Table 3).

As shown in Table 3, after controlling for significantly different clinical characteristics, positive correlations were found between falls and simple reaction time in all participants for the sarcopenia, the frailty, and the sarcopenia and frailty groups (Figure 2).

Despite the theoretical expectations, complex reaction time did not show a significant association with falls, potentially due to a greater variability in the response patterns among participants. A positive statistically significant relationship was found between falls and the number of mistakes made during the complex reaction time in all participant and sarcopenia groups. In the sarcopenia, the frailty, and the sarcopenia and frailty groups negative statistically significant relationships were found between the falls and hand movement frequency. No associations were found between the falls and speed parameters in the control group.

When analyzing the relationship between falls and psychomotor speed parameters between genders, it was found that no associations existed in the men’s group. However, multiple correlations were found in the women’s group (Table 4). In the sarcopenia and the sarcopenia and frailty groups, falls were related to all speed parameters (Figure 3). In the frailty group, only the complex reaction time was not associated with falls. No associations between the falls and the speed parameters were found in the control group.

Multivariable binary logistic regression assessed which psychomotor speed parameters were associated with fall risk. In all participants, a slower simple reaction time (OR: 1.01 [1.002, 1.017], *p* = 0.011) and mistakes made during complex reaction time (OR: 2.449 [1.267, 4.733], *p* = 0.008) were associated with an increased risk of falling. Further analysis showed that only in the sarcopenia group, simple reaction time, as well as mistakes made during complex reaction time, were associated with an increased risk of falling (Table 5).

## 4. Discussion

The results of this study show that falls are related to psychomotor speed parameters, such as reaction time and hand movement frequency in older adults with sarcopenia and frailty. Moreover, these findings are even more emphasized in older women with sarcopenia and frailty. Furthermore, a slower simple reaction time increases the risk of falls in older adults, especially with sarcopenia.

Firstly, we found that the psychomotor speed parameters are related to an increased risk of falling in older adults. It is known that with the aging process, there are changes in cognitive abilities. One of those abilities is psychomotor speed, which tends to slow down with age [25]. Psychomotor speed is an important factor in fall risk, as it encompasses cognitive and motor aspects of safe walking. Previous studies have also found that a slower psychomotor speed or its components are associated with an increased risk of falling, even though methods to assess psychomotor speed were different compared to our study [26,27].

Secondly, in this study, we also found that the relationship between psychomotor speed parameters and falls exists in older adults with sarcopenia and frailty. The information about the relationship between falls and psychomotor speed in older adults with sarcopenia and frailty in previous studies is scarce. A study by Pereira da Silva Alves and colleagues in 2023 also found that slower reaction times in older women with sarcopenia were associated with an increased risk of falling [28]. Slower psychomotor speeds and an increased risk of falls in older adults with sarcopenia could be explained by reduced muscle strength and mass, as they are key criteria for diagnosing sarcopenia [23]. Muscle weakness (especially in the lower limbs) related to sarcopenia leads to poor balance control, slower reaction times, and increased instability when walking or standing. This significantly increases the risk of falling and sustaining injuries like fractures [29]. To our knowledge, no previous study has analyzed the relationship between psychomotor speed and falls in older adults with both sarcopenia and frailty. We found that simple reaction time and hand movement frequency are related to an increased risk of falling. These findings could be explained by decreased muscle strength and physical performance, as both are measured in sarcopenia and frailty [23,24]. In addition, a cognitive component could be a partial factor in these results. Cognitive frailty is a term used to describe frailty coexisting with cognitive impairment [30]. Although our study participants did not qualify for cognitive impairment defined by the cognitive frailty definition, it is possible that an underlying slowing in cognitive processing speed could have an impact on these results. It is worth noting that our findings revealed that there was no relationship between falls and psychomotor speed parameters in healthy older adults. This could be justified by better motor control, movement coordination, and a faster processing speed [31,32]. What is more, our study found that no associations were found between falls and psychomotor speed in the men’s group. The number of men in this study was small, which could explain why no statistically significant results were found. On the other hand, in the women’s group, even in the sarcopenia and frailty group, significant correlations were found between falls and psychomotor speed parameters. A previous study reported that a slower reaction time is associated with an increased risk of falls in women with sarcopenia [28].

Thirdly, our study showed that a slower simple reaction time increases the risk of falling in older adults, particularly in people with sarcopenia. With increasing age, reaction time tends to be slower or variable in older adults [33]. Graveson and colleagues, in the systematic review and meta-analysis, found that a greater variability in reaction time was associated with an increased risk of falling [34]. This could be explained by the reduction in cognitive functions with old age. Because we also found a significant association in the sarcopenia group, muscle strength could be related to a slower reaction time in this population. In addition, sarcopenia is associated with reduced neuromuscular coordination, leading to delayed reaction times and compromised balance control, which likely contributes to fall susceptibility [35]. Previous studies have confirmed that a faster reaction time is associated with increased muscle strength [36,37].

In this study, we found that multiple factors, such as multimorbidity, polypharmacy, handgrip strength, physical performance, physical activity, activities of daily living, and body composition, have an impact on psychomotor speed and falls. These factors were related to falls, psychomotor speed, or both. The effect of multimorbidity and polypharmacy is known to be associated with an increased risk of falling [38]. Lower handgrip strength (a proxy measure of muscle strength) is related to an increased risk of falling and lower psychomotor speed [39,40]. Increased physical performance and physical activity are associated with a reduced risk of falling and faster reaction time [41,42]. Activities of daily living are related to cognitive functioning and psychomotor speed [43]. Psychomotor speed is an essential factor in everyday tasks, which is why a slower psychomotor speed could negatively impact quality of life. Body composition affects falls and psychomotor speed. A lower lean mass and higher fat mass percentage are associated with falls, while a lower appendicular lean mass is associated with a slower psychomotor speed [44,45].

Falls are one of the significant consequences of frailty and can lead to disability, hospitalization, and loss of independence [29]. Sarcopenia is a key component of frailty—muscle loss contributes to weakness and instability, making falls more likely. Both syndromes worsen functional decline, making recovery from falls slower and more difficult. Recurrent falls increase the risk of fear of falling, which leads to less mobility and further muscle loss, creating a closed cycle [46]. Falls are a major consequence of both conditions, leading to disability and loss of independence. Interventions targeting psychomotor speed, such as reaction time training and neuromuscular exercises, may help reduce fall risk in older adults with sarcopenia. Furthermore, preventative strategies (exercise, nutrition, environmental modifications) are essential to reduce fall risk [47].

Our study had both strengths and limitations. We had quite a high number of participants in both the sarcopenia and frailty groups. However, the combined sarcopenia and frailty group was not as large. Furthermore, there was a significant difference between men and women in this study. Although this was a cross-sectional study and a causal relationship is not possible to determine, the findings of this study could help further studies analyze the association between psychomotor speed and fall risk, especially in older adults with sarcopenia and frailty, as their population will grow and health-related issues associated with these disorders will be more prevalent. Longitudinal research could explore whether targeted interventions can improve psychomotor speed and subsequently reduce fall incidences in older adults with sarcopenia and frailty.

## 5. Conclusions

The results of this study show that there is a relationship between falls and psychomotor speed parameters in older adults, particularly with sarcopenia and frailty. In older women with sarcopenia and frailty, falls were related to slower reaction times and hand movement frequency. A slower reaction time was associated with an increased risk of falling in older adults, especially with sarcopenia. After evaluating our results, it is appropriate to investigate the effectiveness of interventions or exercises aimed at improving psychomotor function in reducing the risk of falls in older adults with sarcopenia and frailty.

## Figures and Tables

**Figure 1 medicina-61-00706-f001:**
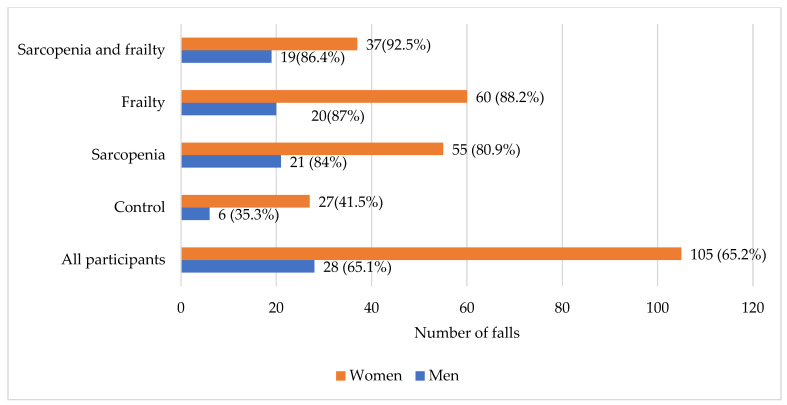
Number of falls in men and women according to sarcopenia and frailty status.

**Figure 2 medicina-61-00706-f002:**
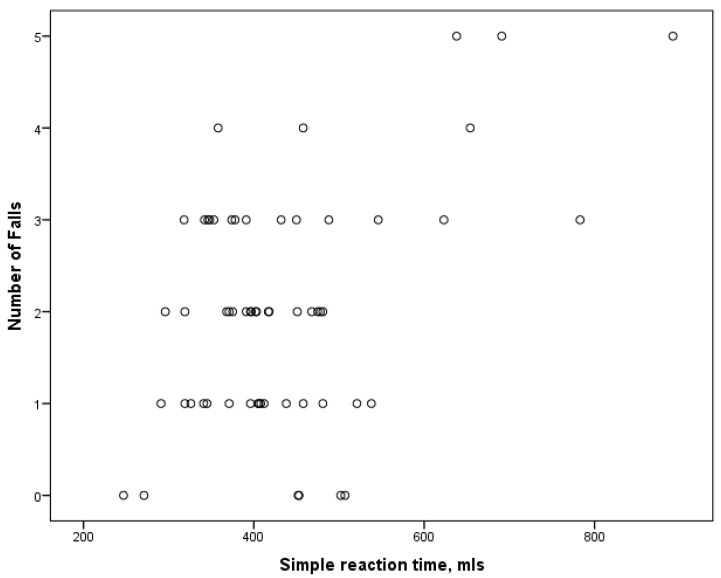
Relationship between falls and simple reaction time in sarcopenia and frailty groups.

**Figure 3 medicina-61-00706-f003:**
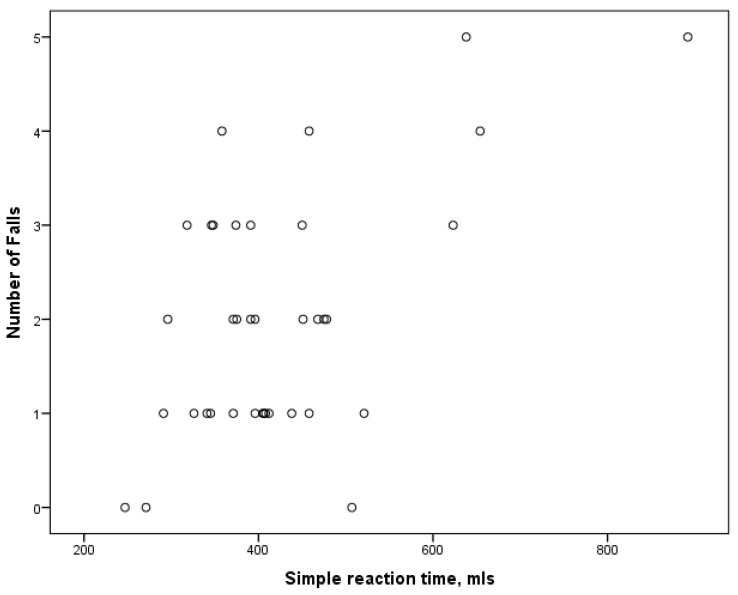
Relationship between falls and simple reaction time in women with sarcopenia and frailty.

**Table 1 medicina-61-00706-t001:** Basic descriptive characteristics of the study population median (25th–75th percentile).

Characteristics	All Participants (n = 204)	Control (n = 82)	Sarcopenia (n = 93)	Frailty (n = 91)	Sarcopenia and Frailty (n = 62)	*p*-Value
Age, years	83 (77–87)	82 (79–84.25)	85 (81–88.5)	85 (82–89)	86 (82–89.25)	<0.001
Female, number (%)	161 (78.9)	65 (79.3)	68 (73.1)	68 (74.7)	40 (64.5)	0.001
Diseases, number	7 (4–9)	4 (3–6.25)	8 (6–9.5)	8 (6–10)	8 (6–10)	<0.001
Medications, number	6 (4–8)	5 (3–7)	6 (4–8.5)	6 (4–8)	6 (3.75–8)	0.536
Polypharmacy (%)	113 (55.4)	37 (45.1)	54 (58.1)	54 (59.3)	32 (51.6)	0.008
Height, cm	167 (163–172)	167.75 (164–172.25)	167 (163–174)	167 (163–174)	169 (163.5–176.72)	0.005
Weight, kg	67.3 (59.7–78.5)	73.3 (64.07–82)	61.4 (55.3–72.6)	63.4 (55.7–78)	61.4 (53.8–77.1)	<0.001
BMI, kg/m^2^	24 (21.59–26.97)	25.24 (23.96–27.6)	21.79 (20.18–24.11)	22.43 (20.34–26.25)	21.55 (19.73–24.71)	<0.001
Dynamometry, kg	14 (12–23.5)	21 (14–29.25)	14 (10–15)	12 (10–16)	12 (9.5–18.5)	<0.001
Gait speed, m/s	0.68 (0.53–0.82)	0.83 (0.71–0.95)	0.59 (0.49–0.76)	0.54 (0.47–0.61)	0.52 (0.47–0.59)	<0.001
SPPB, points	8.5 (6–10)	10 (9–12)	7 (5–9)	6 (4–7)	6 (4–7)	<0.001
PASE, score	35 (27–76)	76 (51–105)	30 (25–48)	27 (25–20)	27 (25–30)	<0.001
ADL, score	6 (4–6)	6 (4–6)	4 (2–6)	4 (2–5)	3 (1–4.25)	<0.001
IADL, score	5 (2–8)	8 (7–8)	2 (2–5)	2 (1–4)	2 (1–2.25)	<0.001
Lean mass, kg	39.01 (32.96–43.31)	41.95 (39.57–46.22)	34.73 (31.09–39.55)	35.52 (31.02–40.04)	34.72 (30.2–39.64)	<0.001
Appendicular lean mass, kg	18.09 (15.31–21.63)	20.97 (18.68–22.93)	15.29 (14.35–17.1)	16.22 (14.66–19.81)	15.51 (14.5–19.53)	<0.001
Fat mass, kg	24.6 (20.18–31.97)	26.97 (22.17–34.24)	21.96 (19.23–29.37)	23.03 (18.63–28.31)	21.64 (18.45–28.44)	0.02

BMI—body mass index; SPPB—Short Physical Performance Battery; ADL—Activities of Daily Living; IADL—Instrumental Activities of Daily Living. The *p*-value was calculated using the Kruskal–Wallis test when comparing control, sarcopenia, frailty, and sarcopenia and frailty groups.

**Table 2 medicina-61-00706-t002:** Falls and psychomotor speed characteristics of study population median (25th–75th percentile).

Characteristics	All Participants (n = 204)	Control (n = 82)	Sarcopenia (n = 93)	Frailty (n = 91)	Sarcopenia and Frailty (n = 62)	*p*-Value
Falls, number	1 (0–2)	0 (0–1)	2 (1–3)	2 (1–3)	2 (1–3)	<0.001
Simple reaction time, mls	326 (285–405)	285 (258–324)	374.5 (318–451)	402 (343.5–479.5)	406 (358–478)	<0.001
Complex reaction time, mls	617 (527–782)	509 (448–578)	697.5 (641.75–873.25)	781 (672–914.5)	781 (673–913)	<0.001
Mistakes of complex reaction, number	1 (0–2)	0 (0–1)	1 (1–2)	1 (1–2)	2 (1–2)	<0.001
10 s tapping test, count	51 (45.75–56)	55 (51–60)	46 (42–51)	46 (41.5–51.5)	44 (41–49)	<0.001
60 s tapping test, count	253 (217–298)	299 (259–323)	228 (194–251.5)	229 (192–268)	215.5 (188–239.75)	<0.001

The *p*-value was calculated using the Krukal–Wallis test when comparing control, sarcopenia, frailty, and sarcopenia and frailty groups.

**Table 3 medicina-61-00706-t003:** Partial correlation coefficient (r) between the falls and the psychomotor speed parameters.

Psychomotor Speed Parameters	Falls				
	All Participants	Control	Sarcopenia	Frailty	Sarcopenia and Frailty
Simple reaction time	0.332 *	0.146	0.42 *	0.486 *	0.444 *
Complex reaction time	0.125	0.116	0.174	0.096	0.118
Mistakes of complex reaction	0.166 *	0.071	0.241 *	0.217	0.206
10 s tapping test	−0.132	−0.047	−0.275 *	−0.277 *	−0.352 *
60 s tapping test	−0.264 *	−0.106	−0.319 *	−0.315 *	−0.312 *

Partial correlation adjusted for age, disease number, polypharmacy, height, weight, BMI, dynamometry, gait speed, SPPB, PASE, ADL, IADL, lean mass, appendicular lean mass, fat mass, and r—zero order (Pearson) coefficient, * *p* < 0.005.

**Table 4 medicina-61-00706-t004:** Partial correlation coefficient (r) between the falls and the psychomotor speed parameters in women.

Psychomotor Speed Parameters	Falls				
	All Participants	Control	Sarcopenia	Frailty	Sarcopenia and Frailty
Simple reaction time	0.331 *	0.202	0.43 *	0.566 *	0.68 *
Complex reaction time	0.141	0.145	0.31 *	0.166	0.406 *
Mistakes of complex reaction	0.221 *	0.083	0.396 *	0.364 *	0.603 *
10 s tapping test	−0.095	−0.025	−0.372 *	−0.291 *	−0.614 *
60 s tapping test	−0.276 *	0.008	−0.467 *	−0.397 *	−0.584 *

Partial correlation adjusted for age, disease number, polypharmacy, height, weight, BMI, dynamometry, gait speed, SPPB, PASE, ADL, IADL, lean mass, appendicular lean mass, fat mass, and r—zero order (Pearson) coefficient, * *p* < 0.005.

**Table 5 medicina-61-00706-t005:** Results of the binary logistic regression for the association between the risk of falls and psychomotor speed in participants with sarcopenia (comparing non-fallers and fallers).

Psychomotor Speed Parameters	OR (95% CI)	*p*-Value
Simple reaction time	1.015 (1, 1.031)	0.048 *
Complex reaction time	0.994 (0.985, 1.002)	0.141
Mistakes of complex reaction	4.067 (1.262, 13.103)	0.019 *
10 s tapping test	1.066 (0.804, 1.413)	0.657
60 s tapping test	0.998 (0.954, 1.045)	0.941

Partial correlation adjusted for age, disease number, polypharmacy, height, weight, BMI, dynamometry, gait speed, SPPB, PASE, ADL, IADL, lean mass, appendicular lean mass, fat mass, * *p* < 0.005.

## Data Availability

The original contributions presented in this study are included in the article. Further inquiries can be directed to the corresponding author.

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
