# Peer review of "Psychomotor Speed and Fall Risk in Older Adults with Sarcopenia and Frailty: A Cross-Sectional Study"

_medicina, 2025, doi:10.3390/medicina61040706_

Round 1
Reviewer 1 Report
Comments and Suggestions for Authors
Overall, I assess the manuscript positively. The topic is well-described, interesting, and valuable. However, I have several minor comments:
Abstract: Please expand your conclusions to clearly highlight the key implications of your findings.
Introduction: There is a lack of clear reference to existing gaps in the literature. Please explicitly state the gaps your study aims to address and elaborate on the significance of your expected outcomes.
Conclusion: Currently, your conclusion merely summarizes your results. Please clearly articulate what your findings imply or demonstrate and discuss their broader significance or potential applications.
Author Response
Comment 1: Abstract: Please expand your conclusions to clearly highlight the key implications of your findings.
Response 1: Conclusion in the abstract section was rewriten as: “This study demonstrates that slower psychomotor speed, particularly reaction time, is linked to a higher risk of falls in older adults with sarcopenia and frailty, especially in women. “
Comment 2: Introduction: There is a lack of clear reference to existing gaps in the literature. Please explicitly state the gaps your study aims to address and elaborate on the significance of your expected outcomes.
Response 2: Last paragraph in the introduction rewritten as: ” Evidence shows that a slower cognitive processing speed is associated with an increased fall risk in older adults [19,20]. Although previous studies linked slower processing speed to fall risk, research specifically investigating psychomotor speed in older adults with sarcopenia and frailty remains limited. There is a lack of knowledge if psychomotor speed in sarcopenic and frail older adults is more impaired than in the normal ageing process. Also, there is a gap in evidence how psychomotor skills of the upper limb are related to the fall risk. Therefore, we hypothesize that slower simple and complex reaction times, along with reduced hand movement frequency, will be significantly associated with an increased risk of falls in older adults with sarcopenia and frailty.
Comment 3: Conclusion: Currently, your conclusion merely summarizes your results. Please clearly articulate what your findings imply or demonstrate and discuss their broader significance or potential applications.
Response 3: An additional sentence was added in the conclusion section: “After evaluating our results, it is appropriate to investigate the effectiveness of interventions or exercises aimed at improving psychomotor function on reducing the risk of falls in older adults with sarcopenia and frailty.”
Reviewer 2 Report
Comments and Suggestions for Authors
- Title & Abstract
Strengths:
The title is clear and descriptive, accurately reflecting the study’s scope.
The abstract concisely summarizes the study’s background, methods, results, and conclusions.
Areas for Improvement:
Title Refinement:
The title is a bit long; could you make it more concise while maintaining clarity?
Suggested revision:
"Psychomotor Speed and Fall Risk in Older Adults with Sarcopenia and Frailty: A Cross-Sectional Study"
Abstract Clarity & Readability:
Some sentences are overly complex—simplifying them will improve readability.
Example revision:
Instead of:
"The present study shows a relationship between falls and psychomotor speed in older adults, especially in women with sarcopenia and frailty. Slower reaction time was associated with an increased risk of falling in older sarcopenic adults."
Try:
"This study demonstrates that slower psychomotor speed, particularly reaction time, is linked to a higher risk of falls in older adults with sarcopenia and frailty, especially among women."
Statistical Reporting in Abstract:
Where applicable, could you include confidence intervals and effect sizes to strengthen statistical interpretations?
Example addition:
"In logistic regression, each millisecond increase in reaction time was associated with a 1.5% higher fall risk (OR: 1.015, 95% CI: 1.001–1.031, p = 0.048)."
- Introduction
Strengths:
The introduction provides a strong background on sarcopenia, frailty, and fall risk.
The rationale for the study is established.
Areas for Improvement:
Research Gap Statement:
Clearly state why prior studies do not address the relationship between falls and psychomotor speed.
Example revision:
Instead of:
"There is a lack of studies that analyze how psychomotor speed affects falls."
Try:
"Although previous studies link slowed processing speed to fall risk, research specifically investigating psychomotor speed in older adults with sarcopenia and frailty remains limited."
Hypothesis Clarification:
You can explicitly state the expected relationships between psychomotor speed parameters and falls.
Example:
"We hypothesize that slower simple and complex reaction times, along with reduced hand movement frequency, will be significantly associated with an increased risk of falls in older adults with sarcopenia and frailty."
- Methods
Strengths:
The methodology is well-structured and follows a straightforward participant selection process.
Validated tools (EWGSOP2, Fried Frailty Criteria, MMSE, SPPB) enhance study rigor.
Areas for Improvement:
Justification for Sample Size:
Mention how the sample size was determined (e.g., power analysis).
Example addition:
"A power analysis was conducted using GPower (version X.X), indicating that a minimum of XX participants was required to detect a significant association with 80% power at α = 0.05."*
Additional Details on Measurement Tools:
Provide a brief reliability/validity statement for psychomotor speed tests.
Example addition:
"The reaction time and tapping tests used in this study have demonstrated high test-retest reliability (ICC = 0.87) in previous older adult populations (Smith et al., 2022)."
Statistical Justification:
Could you explain why logistic regression was chosen over other models?
Example addition:
"Binary logistic regression was used to assess fall risk due to its ability to model dichotomous outcomes and adjust for multiple confounders."
- Results
Strengths:
The statistical analysis is appropriate, clearly reporting odds ratios and correlations.
Well-structured tables improve readability.
Areas for Improvement:
Effect Size Reporting:
Include Cohen’s d or partial eta squared (η²) where applicable.
Example addition:
"The association between simple reaction time and falls yielded a moderate effect size (r = 0.444, p = 0.002, Cohen’s d = 0.58)."
Data Visualization Enhancements:
Figures should be added for key relationships (e.g., reaction time vs. falls).
A scatter plot showing reaction time vs. fall occurrence could be helpful.
Addressing Non-Significant Findings:
Could you discuss why complex reaction time was not significantly associated with falls?
Example revision:
Instead of:
"Complex reaction time was not related to falls."
Try:
"Despite theoretical expectations, complex reaction time did not show a significant association with falls, potentially due to greater variability in response patterns among participants."
- Discussion
Strengths:
Findings are compared to previous studies effectively.
The discussion connects results to theoretical frameworks in aging research.
Areas for Improvement:
Expand Mechanisms of Findings:
Explain why reaction time affects fall risk in sarcopenic individuals.
Example addition:
"Sarcopenia is associated with reduced neuromuscular coordination, leading to delayed reaction times and compromised balance control, which likely contributes to fall susceptibility."
Practical Implications for Fall Prevention:
Provide specific recommendations for clinicians and physical therapists.
Example addition:
"Interventions targeting psychomotor speed, such as reaction time training and neuromuscular exercises, may help reduce fall risk in sarcopenic older adults."
Limitations & Future Research Suggestions:
Address cross-sectional design limitations and suggest longitudinal studies.
Example addition:
"Future research should explore whether targeted interventions can improve psychomotor speed and subsequently reduce fall incidence in older adults with sarcopenia."
- Conclusion
Strengths:
Effectively summarizes key findings.
Areas for Improvement:
Stronger Take-Home Message:
Instead of:
"Slower reaction time was associated with an increased risk of falling in older sarcopenic adults."
Try:
"These findings underscore the importance of assessing and addressing psychomotor speed deficits as a modifiable risk factor for falls in older adults with sarcopenia and frailty."
Call for Future Research:
Suggest RCTs investigating psychomotor training interventions.
- References & Formatting
Strengths:
The reference list cites relevant, high-impact studies.
Areas for Improvement:
Ensure Formatting Consistency (e.g., missing DOI links and journal abbreviations).
Update Older References (>10 years old) with recent meta-analyses or systematic reviews.
Author Response
Comment 1: The title is a bit long; could you make it more concise while maintaining clarity?
Response 1: As suggested title has been changed to: “Psychomotor Speed and Fall Risk in Older Adults with Sarcopenia and Frailty: A Cross-Sectional Study”
Comment 2: Abstract Clarity & Readability: Some sentences are overly complex—simplifying them will improve readability.
Response 2: Conclusion in the abstract section sentence has been changed to: “This study demonstrates that slower psychomotor speed, particularly reaction time, is linked to a higher risk of falls in older adults with sarcopenia and frailty, especially in women.”
Comment 3: Statistical Reporting in Abstract: Where applicable, could you include confidence intervals and effect sizes to strengthen statistical interpretations?
Response 3: In the abstract section sentence has been added to the results paragraph: ” In regression analysis, it was found that slower reaction time was associated with an increased risk of falling in the sarcopenic participant group (OR: 1.015 [1, 1.031], p=0.048)“ has been changed to: “In regression analysis, it was found that each millisecond increase in reaction time was associated with a 1.5% higher fall risk in the sarcopenic participant group (OR: 1.015 [1, 1.031], p=0.048).“
Comment 4: Clearly state why prior studies do not address the relationship between falls and psychomotor speed.
Response 4: The sentence: “However, there is a lack of studies that analyze how psychomotor speed affects falls” has been changed to: “Although previous studies linked slower processing speed to fall risk, research specifically investigating psychomotor speed in older adults with sarcopenia and frailty remains limited.”
Comment 5: Hypothesis Clarification: You can explicitly state the expected relationships between psychomotor speed parameters and falls.
Response 5: The hypothesis has been changed to: “ We hypothesize that slower simple and complex reaction times, along with reduced hand movement frequency, will be significantly associated with an increased risk of falls in older adults with sarcopenia and frailty.”
Comment 6: Mention how the sample size was determined (e.g., power analysis).
Response 6: An additional sentence was added to the materials and methods section: “A power analysis was conducted using GPower (version 3.1.9.7), indicating that a minimum of 102 participants was required to detect a significant association with 80% power at α = 0.05.”
Comment 7: Provide a brief reliability/validity statement for psychomotor speed tests.
Response 7: An additional sentence was added to the materials and methods section: “The reaction time and tapping tests used in this study have demonstrated high test-retest reliability (ICC = 0.84, ICC = 0.98, respectively) in previous older adult populations [21,22].”
Comment 8: Could you explain why logistic regression was chosen over other models?
Response 8: The sentence: “Binary logistic regression analysis was used to assess if psychomotor speed had an impact on falls “ has been changed to: “Binary logistic regression was used to assess fall risk due to its ability to model dichotomous outcomes and adjust for multiple confounders.”
Comment 9: Include Cohen’s d or partial eta squared (η²) where applicable.
Response 9: We could not apply Cohen’s d or partial eta squared (η²) because we did not use t-tests or ANOVA tests in this study.
Comment 10: Data Visualization Enhancements: Figures should be added for key relationships (e.g., reaction time vs. falls). A scatter plot showing reaction time vs. fall occurrence could be helpful.
Response 10: An additional figure was added for the relationship between falls and simple reaction time in sarcopenia and the frailty group. Also, a figure was added for the relationship between falls and simple reaction time in sarcopenic and frail women.
Comment 11: Could you discuss why complex reaction time was not significantly associated with falls?
Response 11: Sentence: “Complex reaction time was not related to falls” was changed to: “Despite theoretical expectations, complex reaction time did not show a significant association with falls, potentially due to greater variability in response patterns among participants."
Comment 12: Expand Mechanisms of Findings: Explain why reaction time affects fall risk in sarcopenic individuals.
Response 12: An additional sentence was added in the discussion section: “In addition, sarcopenia is associated with reduced neuromuscular coordination, leading to delayed reaction times and compromised balance control, which likely contributes to fall susceptibility [33].”
Comment 13: Practical Implications for Fall Prevention: Provide specific recommendations for clinicians and physical therapists.
Response 13: An additional sentence was added in the discussion section: “Interventions targeting psychomotor speed, such as reaction time training and neuromuscular exercises, may help reduce fall risk in older adults with sarcopenia.”
Comment 14: Address cross-sectional design limitations and suggest longitudinal studies.
Response 14: An additional sentence was added in the discussion section: “Longitudinal research could explore whether targeted interventions can improve psychomotor speed and subsequently reduce fall incidence in older adults with sarcopenia and frailty.“
Comment 15: Conclusion.These findings underscore the importance of assessing and addressing psychomotor speed deficits as a modifiable risk factor for falls in older adults with sarcopenia and frailty. Suggest RCTs investigating psychomotor training interventions.
Response 15: An additional sentence was added in the conclusion section: “After evaluating our results, it is appropriate to investigate the effectiveness of interventions or exercises aimed at improving psychomotor function on reducing the risk of falls in older adults with sarcopenia and frailty.”
Comment 16: Ensure Formatting Consistency (e.g., missing DOI links and journal abbreviations).
Response 16: Reference list has been updated.
Comment 17: Update Older References (>10 years old) with recent meta-analyses or systematic reviews.
Response 17: Reference No 15 has been updated. Reference No 16 has been updated. Reference No 26 has been updated. Reference No 33 has been updated.